# Transition from sexuality to androgenesis through a meiotic modification during spermatogenesis in freshwater *Corbicula* clams

Emilie Etoundi[1], Martin Vastrade[1]*, Clothilde Berthelin[2], Kristell Kellner[2], Mélanie Fafin-Lefèvre[3], Karine Van Doninck[1,4]*

1 Laboratory of Evolutionary Genetics and Ecology, Research Unit in Environmental and Evolutionary Biology, Institute of Life, Earth, and Environment, University of Namur, Namur, Belgium, 2 Unité Biologie des Organismes et des Ecosytèmes Aquatiques (BOREA, UMR 7208), Université de Caen Normandie, Sorbonne Université, Museum National d'histoire Naturelle, Université Pierre et Marie Curie, CNRS, IRD, Caen, France, 3 INSERM U1086 "Cancers et Préventions", Centre François Baclesse, Caen, France, 4 Université libre de Bruxelles (ULB), Molecular Biology & Evolution, Brussels, Belgium

* karine.van.doninck@ulb.be (KVD); martin.vastrade@unamur.be (MV)

**Data Availability Statement:** All relevant data are within the manuscript and its Supporting Information files.

## Abstract

Asexual taxa are often considered as rare and vowed to long-term extinction, notably because of their reduced ability for rapid genetic changes and potential adaptation. The rate at which they derive from sexual ancestors and their developmental mode however influence genetic variation in asexual populations. Understanding the transition from sexuality to asexuality is therefore important to infer the evolutionary outcome of asexual taxa. The present work explored the transition from sexuality to androgenesis, a reproductive mode in which the males use female resources to clone themselves, in the freshwater *Corbicula* clams. Since androgenetic lineages are distinguishable from sexual clams by the production of unreduced sperm, this study investigated the cytological mechanisms underlying spermatogenesis in *Corbicula* by following the DNA content variation of male germ cells. The widespread androgenetic *C.* sp. form A/R lineage was compared to the sexual species *C. japonica* and *C. sandai*. While in *C. japonica*, the last stages of spermatogenesis are reduced through a canonical meiosis process, no reduced or duplicated stages were observed in *C.* sp. form A/R, suggesting a meiosis modification in this lineage. However, 45% of *C. sandai* spermatozoa were unreduced. The production of unreduced sperm may condition or provide the potential for the emergence of androgenesis in this sexual species. Being closely related to androgenetic lineages and found in sympatry with them in Lake Biwa (Japan), *C. sandai* might be an origin of androgenetic lineage emergence, or even an origin of the androgenetic reproductive mode in *Corbicula*.

## Introduction

Sexual reproduction *sensu stricto* refers to meiotic sex involving a two-step process during which genomes are diversified by a nuclear division called meiosis [1], whereby recombination

**Funding:** The present study has been funded by the University of Namur and the National Funds for Scientific Research (FNRS) through a PhD thesis grant (FC91712) to E.E. and by a PDR research grant (CARMA, 14596412) to K.V.D. The funders had no role in study design, data collection and analysis, decision to publish, or preparation of the manuscript.

**Competing interests:** The authors have declared that no competing interests exist.

occurs between homologous chromosomes, followed by a nuclear fusion between haploid maternal and paternal gametes [1]. Meiotic sex is an ancient eukaryotic innovation, being almost universal in higher animals and plants [2]. Its maintenance and near ubiquity indicate how important it is for life. Despite this, various asexual reproductive modes have arisen independently and repeatedly within very distinct taxa, and the term "asexuality" refers to a panoply of unrelated reproductive modes [3, 4]. Among this diversity of reproductive systems, some are considered asexual because of the transmission of only one of the parental genomes but still require the occurrence of both male and female gonads producing spermatozoa and oocytes [5]. One such reproductive mode is androgenesis, in which the offspring only inherits the entire paternal nuclear genome [reviewed in [6]]. It is considered as sexual parasitism, where female host resources are used by the males to spread their genes [5].

Spontaneous androgenesis was reported in some unrelated taxa such as ants [*Wasmannia auropunctata* [7]; *Vollenhovia emeryi* [8]; *Paratrechina longicornis* [9]; *Cardiocondyla kagutsuchi* [10]], bees [*Apis mellifera* [11]], stick insects [*Bacillus* hybrids [12]; *Leptynia hispanica* [13]; *Clonopsis* hybrids [14, 15]; *Pijnackeria* hybrids [16]], clams [*Corbicula* spp. [17]] and one tree [*Cupressus dupreziana* [18]]. Spontaneous androgenesis was also hypothesized to occur in the wasp *Venturia canescens*, in several plant hybrids [6] and in the fish *Poecilia formosa*, based on some ancient observations [19, 20] but the genetic evidences are still lacking. However, an androgenetic reproductive mode is not easily detectable because genetic information is often required, and its actual occurrences within the tree of life could be underestimated [21].

In the androgenetic systems reported, the cytological mechanisms causing the maternal nuclear genome loss and, in some cases, the production of unreduced male gametes, are often poorly known. In ant species, androgenesis usually co-occurs with parthenogenesis, the development of an individual from an unreduced egg [22]. Several hypotheses were given to explain the maternal genome loss in ants, such as its elimination or destruction driven by the father [7, 23] or following chromosome missegregation [22], or the production of anucleate ovules as a by-product of thelytokous ovule formation [23]. Given the haplodiploid system of ants, haploid sperm formed during androgenesis will form viable haploid males [24], unreduced sperm are not required in this system.

In the stick insect species studied, androgenesis is often associated with hybridogenesis, and the maternal genome loss would be due to an entire egg nucleus degeneration and expulsion as polar bodies [25, 26]. Two distinct androgenetic systems were reported coexisting. One produces a bisexual offspring, often biased towards females, with a heterozygous genetic structure that would only be explained by the fusion of two distinct spermatozoa with the oocyte. Another one produces an all-male progeny which seems genetically identical to the father, suggesting the production of unreduced sperm that transmits its entire nuclear genome to the descendants [27]. These two mechanisms (ploidy restoration through spermatozoa fusion and ploidy maintenance via unreduced sperm production) could thus co-occur in stick insects [27].

In the African conifer *Cupressus dupreziana*, reproducing through obligate androgenesis, the somatic DNA content is maintained through the production of unreduced pollen [28]. The female gametes do not participate to the offspring formation as the male gametes only need the female endosperm–vegetative resources next to the oocyte but that are not part of it–to develop into an embryo, without fertilization [18]. Interestingly, the mitochondria and chloroplast DNA is also paternally inherited in Cupressaceae family [29, 30], confirming the possibility for an embryo to develop without a female gamete. In France, where it was introduced, *C. dupreziana* can also reproduce using the endosperm of a related species, the Mediterranean conifer *C. sempervirens* [31]. Noteworthy, even though *C. sempervirens* produces reduced pollen, it is also capable of androgenetic reproduction using the endosperm of *C. dupreziana*,

resulting either in viable haploid offspring, or diploid offspring through an unknown ploidy restoration mechanism (gamete fusion or diploidization) [32]. Again, the two mechanisms (unreduced pollen production and pollen fusion or diploidization) appear to co-occur in *Cupressus* conifers. The female gamete formation and outcome (putative development or abortion) are still poorly known in this system.

Finally, the notorious clade in which androgenesis is widespread among several species and best characterized is the animal clam genus *Corbicula* [17, 33, 34]. This genus includes both sexual and androgenetic species [35]. The sexual species are reported as dioecious with males producing uniflagellate haploid sperm, and parental ploidy being restored in the offspring through cross-fertilisation [36–38]. The androgenetic *Corbicula* lineages, however, produce biflagellate unreduced spermatozoa [36, 37, 39] and are reported as hermaphrodites [[37, 40] but see [41, 42]]. After fertilization of the oocyte by this unreduced sperm (both cross- and self-fertilizations occur), all maternal nuclear chromosomes are expulsed as two polar bodies. This occurs during the first meiotic division following fertilization, while the second meiotic division was not observed [17, 43]. As a result, in androgenetic *Corbicula* lineages only the paternal nuclear chromosomes are transmitted to the offspring together with the maternal mitochondrial genome.

The cytological mechanisms leading to unreduced sperm formation in androgenetic *Corbicula* clams are not understood. Few sperm or histological investigations were carried out in this androgenetic system, notably reporting the monoflagellate morphology of the sexual *C. sandai* spermatozoa [37] and the biflagellate morphology of the spermatozoa from androgenetic lineages [39]. The different cell stages of the spermatozoa differentiation process were described in one androgenetic *Corbicula* lineage [37] reporting four distinct stages, but they were never associated to a DNA content. It has been predicted but not verified that in androgenetic *Corbicula* lineages, one of the meiotic divisions might be abortive during spermatogenesis. Such aborted meiotic division was observed during oogenesis [36].

Since hermaphroditic androgenetic and dioecious sexual lineages co-exist in the clam genus *Corbicula*, it represents a suitable animal model system to understand the proximate causes of transitions from sexuality to androgenesis and how unreduced male gametes are formed. Remarkably, in Lake Biwa in Japan, sexual *Corbicula sandai* and androgenetic *Corbicula* spp. clams co-occur. Previous genetic studies [34, 44–46] also demonstrated that this Lake Biwa population is particularly genetically diverse and includes a large part of the *Corbicula* genetic diversity found worldwide, with *C. sandai* sharing alleles with at least three out of the five invasive androgenetic lineages [46]. Lake Biwa is therefore a hotspot of plausible origin of androgenetic *Corbicula* lineages where transitions from sexual to androgenetic lineages could occur.

Biflagellate and unreduced sperm is a hallmark of androgenesis in *Corbicula*, these two features being always observed together, while sexual species appear to produce reduced, uniflagellate sperm. Our study thus focused specifically on DNA content variations during spermatogenesis in both sexual and androgenetic *Corbicula*. The production of unreduced gametes could have emerged through variable mechanisms to restore or maintain the parental DNA content and ploidy in the zygote [see [47] for a review]. From a DNA content perspective [C-Value, with 1C defined here as the DNA content of an unreplicated reduced chromosome set, see [48]], the distinct modifications from canonical meiosis (Fig 1A) can be broadly classified into three categories: i) a modification during meiosis I or II (Fig 1B), ii) a pre-meiotic modification (Fig 1C) or iii) a post-meiotic modification (Fig 1D). The first type, a modified meiosis, usually involves the suppression of one meiotic division, either the first or the second one, whereby the gamete precursors do not reach a DNA content below 2C (Fig 1B) [3]. In the second type, a premeiotic doubling (or endoreplication) occurs during the S phase [49, 50], a

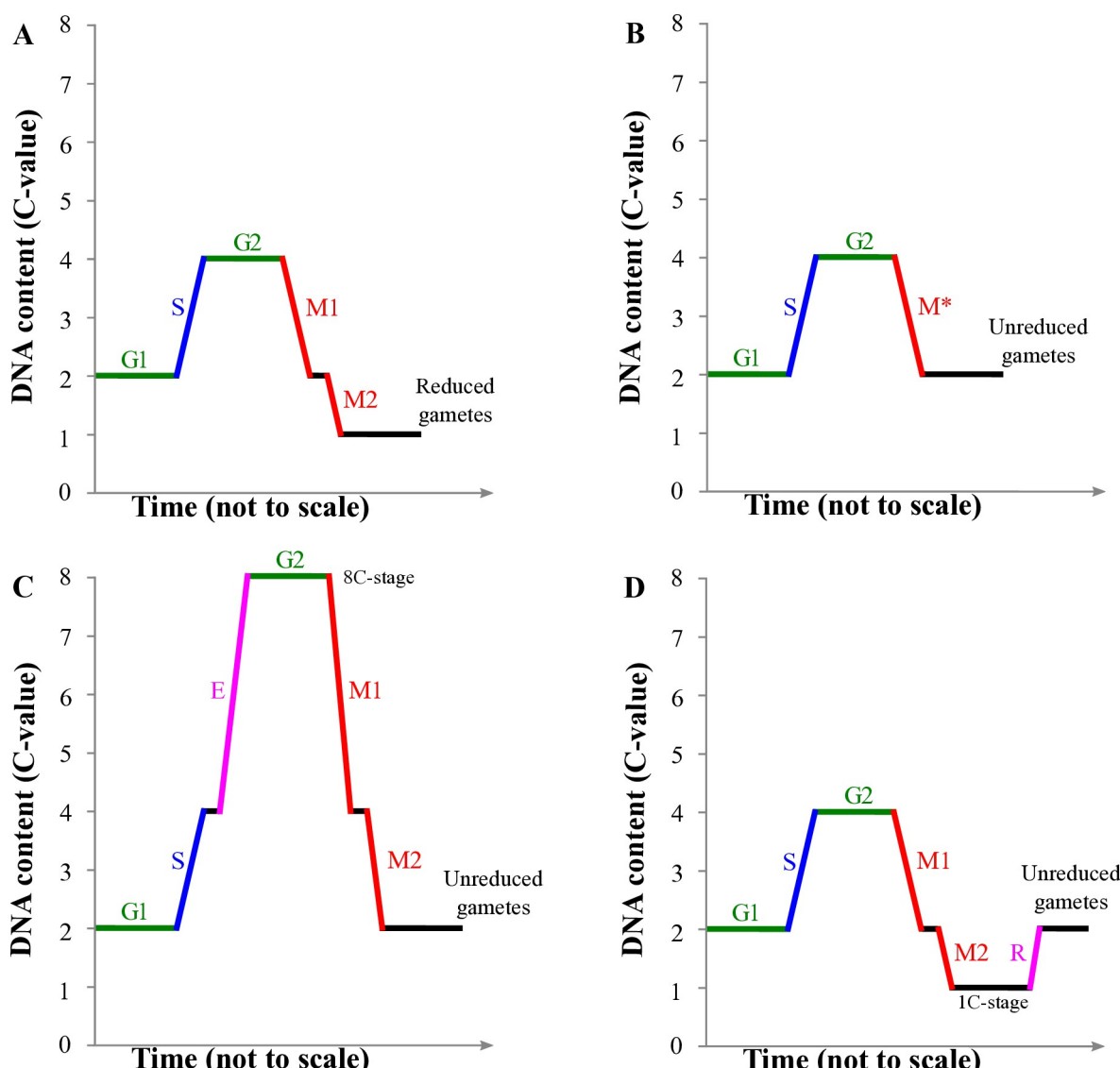

**Fig 1.** Theorical expectation of DNA content variation through the formation of reduced (A) and unreduced (B, C, D) gametes. The C-Value is defined here as the DNA content of an unreplicated reduced chromosome set. The nomenclature G1, S, G2, M1 and M2 refers to the cell cycle: G1 = gap1, the phase between cell division and DNA replication; S = synthesis phase, in which the DNA replication occurs; G2 = gap2, the phase between DNA replication and cell division; M1 and M2 correspond to the two meiotic divisions. **A.** Canonical meiosis, with production of reduced 1C cells. **B.** Modified meiosis with one abortive step (M* in the figure), formation of unreduced gametes without any reduced (1C) stage. **C.** Pre-meiotic endoreplication (E in the figure), formation of unreduced gametes with the occurrence of an 8C-stage, resulting from the endoreplication of a replicated unreduced chromosome set (4C). **D.** Canonical meiosis followed by a post-meiotic endoreplication or gamete fusion, restoration (R in the figure) of the gamete DNA content after a reduced 1C-stage.

process in which the chromosome number is doubled with the DNA content reaching 8C (in diploids) and then reduced via a normal meiotic division to 2C (Fig 1C). Finally, in the third category, a canonical meiosis results in a reduced stage (1C DNA content, *sensu* Greilhuber) followed by a post-meiotic restoration of the parental DNA content through endoreplication or through sperm nuclei fusion (Fig 1D). These three different categories (Fig 1B–1D) do not exhaustively describe the complexity and diversity of mechanisms producing unreduced gametes. It is however legitimate to simplify the distinction using DNA content as a main criterium

(Fig 1B–1D) as it can be measured throughout spermatogenesis and was already used before [*e.g.*, see [51]]. We therefore used here computer-assisted image analysis cytometry combined to DNA content measurements to compare spermatogenesis in androgenetic *Corbicula* form A/R [34] with sexual *Corbicula* clams, in particular the sexual species *C. sandai* endemic to Lake Biwa and *C. japonica* being found in eastern Asian brackish waters [52]. In this study, we describe for the first time the associated DNA content of the different cell stages of the spermatozoa differentiation process, for both sexual and androgenetic *Corbicula* lineages. We hypothesize that the sexual species *C. japonica* and *C. sandai* perform a canonical meiosis (Fig 1A) and therefore three distinct DNA content states (4C, 2C and 1C-stages) should be detected during spermatogenesis, with the mature spermatozoa being haploid. In the androgenetic *C.* sp. form A/R, however, the mature spermatozoa should be unreduced, and the measured DNA content stages should depend on the ploidy restoration/maintenance mechanism (Fig 1B–1D).

## Material and methods

This research did not involve vertebrates or cephalopods; thus, no ethic approval was required. No field collection took place in protected areas, thus no permits were required.

### Specimen collection

Specimens of *Corbicula* sp. form A/R were sampled with a hand net in the Belgian section of the Meuse River in Petit Lanaye. This androgenetic form is the most widespread and abundant in the invasive range [*e.g.*, [53–55]] but also in the native region [*e.g.*, [56, 57]]. Sexual species, used as control, had to be acquired from another continent, because none has ever been recorded in the invaded area. *C. sandai* individuals were kindly provided by the Seta Fishery cooperative which specifically rears this species (Ohtsu City, Shiga, Japan). This species is related to several androgenetic lineages [44, 46] and might serve as a reservoir for new androgenetic species, some of them being introduced in other continents. It is thus the perfect control for form A/R study. Individuals of the sexual lineage *C. japonica* were collected with a hand net in Shiomi River, Huyga-City (Miyazaki, Japan), 6 km upstream from the river mouth. Individuals were brought to the laboratory and kept alive in an oxygenated aquarium filled with dechlorinated tap water and fed with cultured *Chlorella vulgaris* [58] until further processing.

### Histological study of the gametogenesis process of androgenetic Corbicula form A/R

We carried out a histological investigation of *C.* sp. form A/R gametogenesis during one full year to identify when male and female follicles are simultaneously mature. Monthly samples of *C.* sp. form A/R were taken in the Meuse River between February and January. Individuals were divided into two size classes: class 1 with a shell height of 19.77 ± 1.13 mm (n = 60) and class 2 with a shell height of 10.05 ± 1.16 mm (n = 60). Five individuals per month were selected for each size class. All specimens brought back alive to the laboratory were fixed in Davidson fixative at 4°C for 48–72 hours. After Davidson fixation, specimens were dehydrated in serial dilution of ethanol, embedded in paraffin and 4 μm transverse sections were made [59] at a depth ranging from one third to the half of the animal body mass where the highest gonadal density was retrieved. Sections were stained with a Prenant-Gabe trichrome coloration. Slides were observed with a Nikon Eclipse 80i microscope coupled to a Nikon DXM1200-C camera. Measurements of nuclei were made with the Nikon NIS Elements D software. For each slide (1 per individual), all gametogenic tubule regions were identified and

the proportion of each tubule regions type (hermaphrodite, spermatogenic, oogenic) was measured. Male germ cells were characterized based on their nuclear morphology and size to describe the main stages of spermatogenesis depending on the abundance and appearance of each male germ cell type.

Finally, few slides were prepared from mature form A/R individuals sampled end of May when all sperm cell types are present. They were Feulgen stained (see S1 File) to study in more details the male gonad structure and morphology of spermatogenic cells. Feulgen staining specifically and stoechiometrically binds DNA and will allow to measure average DNA content of each cell type through image cytometry [see [60] for more details]. The DAPI staining is less adapted for this kind of analysis.

## Image analysis cytometry and investigation of the spermatogenetic cell's DNA content

Besides *C.* sp. form A/R, *C. sandai* and *C. japonica* individuals, considered strictly dioecious sexual species [34, 61, 62] were added to the present analysis. Both species are expected to perform a canonical meiosis [63] and are used as reference to validate the image cytometry method.

We prepared paraffin blocks of *C. japonica* (n = 20) and *C. sandai* (n = 21), both collected in July in Japan, and of *C.* sp. form A/R (n = 10) collected end of May, following the procedure described previously. The reproductive status (presence of mature gametes and sex) of these individuals was histologically assessed: 4 µm sections, located in an area with high gametogenic follicles density, were stained using haematoxylin and eosin coloration. Out of 20 *C. japonica* clams analysed, 6 were females, 2 immatures (*i.e.*, no mature gametes were found) and from the 12 males only 7 were kept because they contained all male germ cell types for further analyses. Among the *C. sandai* individuals, we found 8 females, one immature clam and 12 males from which we selected 8 individuals with all spermatogenesis stages for image cytometry. From the 10 *C.* sp. form A/R individuals, 9 were mature hermaphroditic animals suitable for image analysis cytometry.

Cell suspension for image cytometry were directly prepared from the paraffin blocks. For each individual, four successive 50 µm sections were pulled together into histological cassettes. Tissue collected in the cassettes was deparaffinised through successive toluene baths followed by decreasing ethanol concentrations until complete rehydration into distilled water. Deparaffinised tissue fragments were then disaggregated in Medicon 35 µm (Becton Dickinson) following the manufacturer's protocol. The resulting cell suspension was filtered on 50 µm filcons (Becton Dickinson) and then resuspended in methanol for storage. For each *Corbicula* lineage, the cell suspensions of all individuals had to be pooled to obtain the proper cell density on slides. Samples for image analysis cytometry were automatically prepared using the Thin-Prep monolayer method (Marlborough, Massachusetts, USA). Slides obtained through this method were stained with the thionin-Feulgen technique (Perceptronix Clear 2C staining kit) according to the manufacturer's protocol. As this stain stoechiometrically binds to DNA, the staining is proportional to the DNA content and the Integrated Optical Density (IOD) can be measured. The IOD of an object is the total optical density of all pixels constituting this object. We also added a somatic reference. Somatic cells were retrieved from the gills (preliminary checked for absence of embryo incubation) and the mantle of three *C.* sp. form A/R individuals, embedded in paraffin and prepared as previously described. This also enabled us to compare the ploidy distribution of somatic and sperm cells in this androgenetic form.

Slides for image cytometry were analysed and image acquisition was performed with X40 magnification on an automatized Zeiss Axioskop microscope equipped with a 600-nm

interferential filter, a black and white Sony CCD camera. Prior to acquisition, an empty field (containing no cells) was acquired to correct light instability, intensity, and inhomogeneity. This field also served to measure incident and transmitted light to estimate IOD. To assess the cell DNA content of germ cells, we sampled 200 microscopic fields on each slide; for somatic cells however, we analysed 600 fields due to cell scarcity. On each slide, an automated focus was performed, with systematic elimination of empty fields.

Cell nuclei observed on the images collected were analysed using the DRACCAR software (Caen, France). First, through a step of segmentation, all objects on an image were delineated. During segmentation, parameters pertaining to nuclear shape, size, texture (comprehensively listed in Fafin-Lefèvre *et al.* 2011) and IOD are measured. The delimited objects were then manually sorted one by one into 7 classes based on the preliminary histological study, including the different male germ cell types (5 classes), somatic cells and debris. Cells that could not be firmly described, like cell aggregates or unidentified objects, fitted into this last category. Finally, the DNA content (= IOD) of each cell was measured using the fluorimetry method previously described [60].

## Results

### Histological description of spermatogenesis in Corbicula sp. form A/R

The histological follow-up of *C.* sp. form A/R confirmed simultaneous hermaphroditism [37, 40–42] with gonadal tubules composed of oogenic, spermatogenic and hermaphroditic regions. The scarcity of mature gametes in *C.* sp. form A/R class 2 individuals revealed their immaturity or early maturity. Moreover, when present, gonads in class 2 individuals are either females (12/60) or simultaneous hermaphrodites (48/60), suggesting that oogenic tissue develop first in *C.* sp. form A/R. In mature individuals (class 1), the percentage of each gametogenic tissue type (oogenic, spermatogenic, hermaphrodite) does not vary much throughout the year, neither the total volume of the gonadal tubules. However, female tissue is always more abundant than male tissue in terms of regions number, but also in volume (S1 Fig).

This histological study also allowed us to determine the different stages of spermatogenesis: six developmental stages of spermatogenesis could be described in *Corbicula* sp. form A/R (S1 Table). These arbitrary subdivisions are used to describe a continuous process, hence transitionary phases may be encountered. Advanced spermatogenesis stages with mature spermatozoa (stages II to IV) are present throughout the whole year, except in March and April (S1 Fig). The presence of ripe follicles actively discharging gametes and the observation of ctenidial incubated-embryos suggest two reproductive peaks: the first in late spring—summer and the second at the end of fall, beginning of winter. From this study, the time points could be determined at which all male germ cell types are present within the gametogenic follicles of androgenetic *C.* sp. form A/R: *i.e.*, late spring-early summer. This time point was therefore chosen for further analyses of spermatogenesis in androgenetic *C.* sp. form A/R.

*Corbicula* gonads are constituted of tubules embedded within connective and storage tissue, which surround the digestive tissue (Fig 2). Spermatogenic regions are mostly located in terminal sections of gametogenic tubules or along the visceral mass margin. They are also characterized by the massive year-round presence of macrophages, indicating considerable lysis. Spermatogenic tubule regions are well structured and show a classical centripetal organization. The outer most cells are stem cells and spermatogonia. Putative stem cells are very uncommon with a hardly stained nucleus and a pale nucleolus, because of completely decondensed chromatin (Fig 2). Rarity of these cells did not allow us to measure enough nuclei to estimate an average nucleus size. Early spermatogonia (5.4 ± 0.69 μm, n = 51) appear as pale spherical or ellipsoid nuclei with a low abundance of heterochromatin; these cells present a dark nucleolus

| Nucleus representation | Cell type | Nucleus diameter (µm) | Nucleus description | |
|---|---|---|---|---|
| | Putative stem cell | Not estimated | Almost transparent, frequent nucleolus | |
| | Spermatogonium | 5,40 ± 0,69 (n=51) | Pale, frequent nucleolus | |
| | Spermatocyte | 3,93 ± 0,41 (n=49) | Dark, punctuated | |
| | Spermatid | 1,71 ± 0,34 (n=75) | Dense | |
| | Spermatozoon | Not estimated here  13,9 ± 0,32 (Konishi *et al*. 1998) | Dense, elongated | |

**Fig 2. Histological description of spermatogenesis in *Corbicula* sp. form A/R: The different cell types observed.** For the general anatomical organization of *Corbicula* clams, a section through a mature individual of *C.* sp. form A/R sampled in May 2012 is represented. CT: connective tissue, DT: digestive tissue, G: gills, M: mantle, NT: nervous tissue, OR: oogenic region of gonadal tubule, HR: hermaphroditic region of gonadal tubule, SR: spermatogenic region of gonadal tissue. Prenant-Gabe Trichrome coloration. A zoom of one SR region enables the identification of the different cell types of spermatogenesis. Spg: Spermatogonium Spc: Spermatocyte Spd: Spermatid and Spz: Spermatozoid.

(Fig 2). Closer to the tubule lumina the spermatocytes are found, grouped into cell clones. Spermatocytes (3.93 ± 0.41µm, n = 49), because of their condensed chromatin content, have dark and punctuated nuclei without nucleoli (Fig 2). No distinction could be made between spermatocytes I and II based on their nuclear aspect in light microscopy. The most central germ cells are spermatids (1.71 ± 0.34 µm, n = 75) and spermatozoa. Both cell types appear as dense nuclei. The spermatid nucleus shape varies according to the spermiogenesis stage, going from round to oblong. Spermatozoa have elongated, conical-shaped nuclei; their flagella are hardly distinguishable (Fig 2). Male germ cells are generally easily distinguishable based on their nuclear morphology, although some stages may have overlapping sizes (Fig 2).

## Image cytometry conducted on androgenetic and sexual Corbicula clams

Based on the histological investigation of spermatogenesis of *C.* sp. form A/R, sperm cell types were classified into five categories for both androgenetic and sexual *Corbicula* individuals: (1.) spermatozoa; (2.) spermatids; (3.) spermatocytes; (4.) spermatogonia; (5.) putative stem cells (Fig 2). Most of the recorded objects were considered as debris (75.4% for *C. sp.* form A/R somatic cells, 82.2% for *C. sp.* form A/R germ cells, 72.4% for *C. japonica* and *C. sandai* germ cells). No putative stem cells were recorded in any of the samples.

Sampled *C. japonica* and *C. sandai* from the Seta fisheries analyzed in this study were dioecious with individuals containing only spermatogenic or oogenic tubules, no hermaphroditic region was observed. In sperm collected and analyzed from *C. japonica* males (n = 7), four distinct cell types could be identified: spermatogonia, spermatocytes, spermatids, and spermatozoa. The IOD content of each cell type was measured with the DRACCAR software, and three main cell ploidy populations could be delimited arbitrarily (Fig 3). Firstly, 53.57% (488/911) of the sorted cells grouped into a 1C population of spermatozoa and spermatids. A 2C population

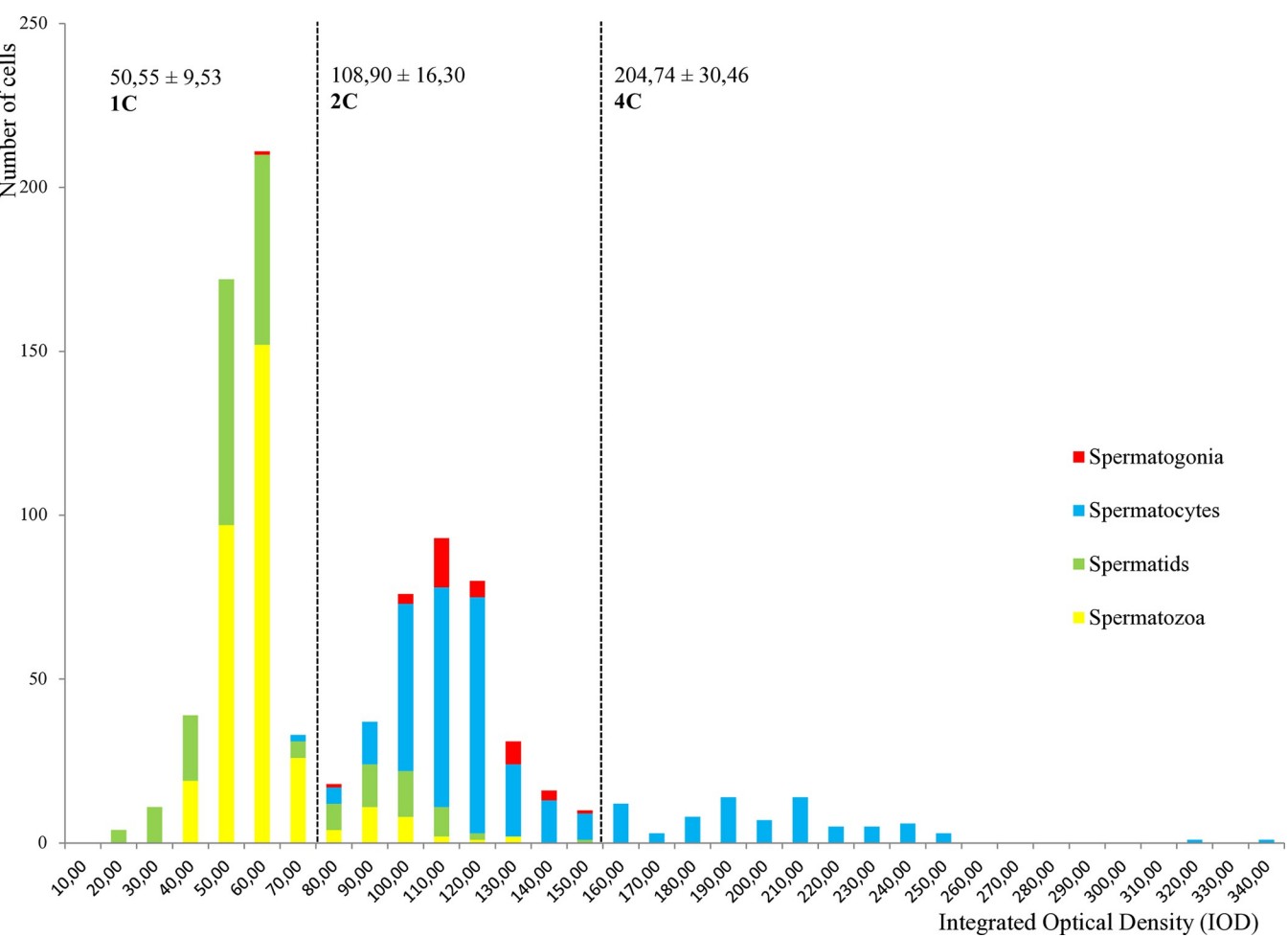

**Fig 3. DNA content variation during spermatogenesis of sexual *C. japonica* (n = 7).** Total number of cells counted: 911. 1C (n = 488), 2C (n = 343), 4C (n = 80) cells, where C is the DNA content expressed as the integrated optical density (IOD) given on the X-axis. For each category, the average IOD ± standard deviation is provided. The number of cells is given on the Y-axis.

mostly constituted of spermatocytes and spermatogonia that represented 37.65% (343/911) of the total cells. Finally, the remaining 8.78% (80/911) of the total cells constituted a 4C population of spermatocytes. It is however worth noting that a small proportion of spermatozoa and spermatids, 8.7 and 15.5% respectively, are found in the 2C cell population and are therefore unreduced. They have an average Integrated Optical Density (IOD) of 92.56 ± 13.64.

Four cell types were also detected in sperm from *C. sandai* males (n = 8), and they could also be categorized into three ploidy populations arbitrarily (Fig 4). The 1C (44.72% of cells, 233/521) population was composed of spermatids and spermatozoa, the 4C (7.10% of cells, 37/521) population mainly of spermatocytes as in *C. japonica*. Surprisingly, all cell types were found in the 2C population (48.18% of cells, 251/521), including spermatozoa (Fig 4). A total of 44.76% of all the spermatozoa identified and 33.33% of all spermatids are found to be unreduced and present in the 2C population of *C. sandai* (average IOD: 100.71 ± 12.26).

All form A/R individuals investigated are hermaphrodites and their DNA content distribution is represented in an IOD frequency histogram, both for somatic and germ cells (Fig 5). While distinct ploidy populations could not be retrieved as clearly as for *C. sandai* and *C.*

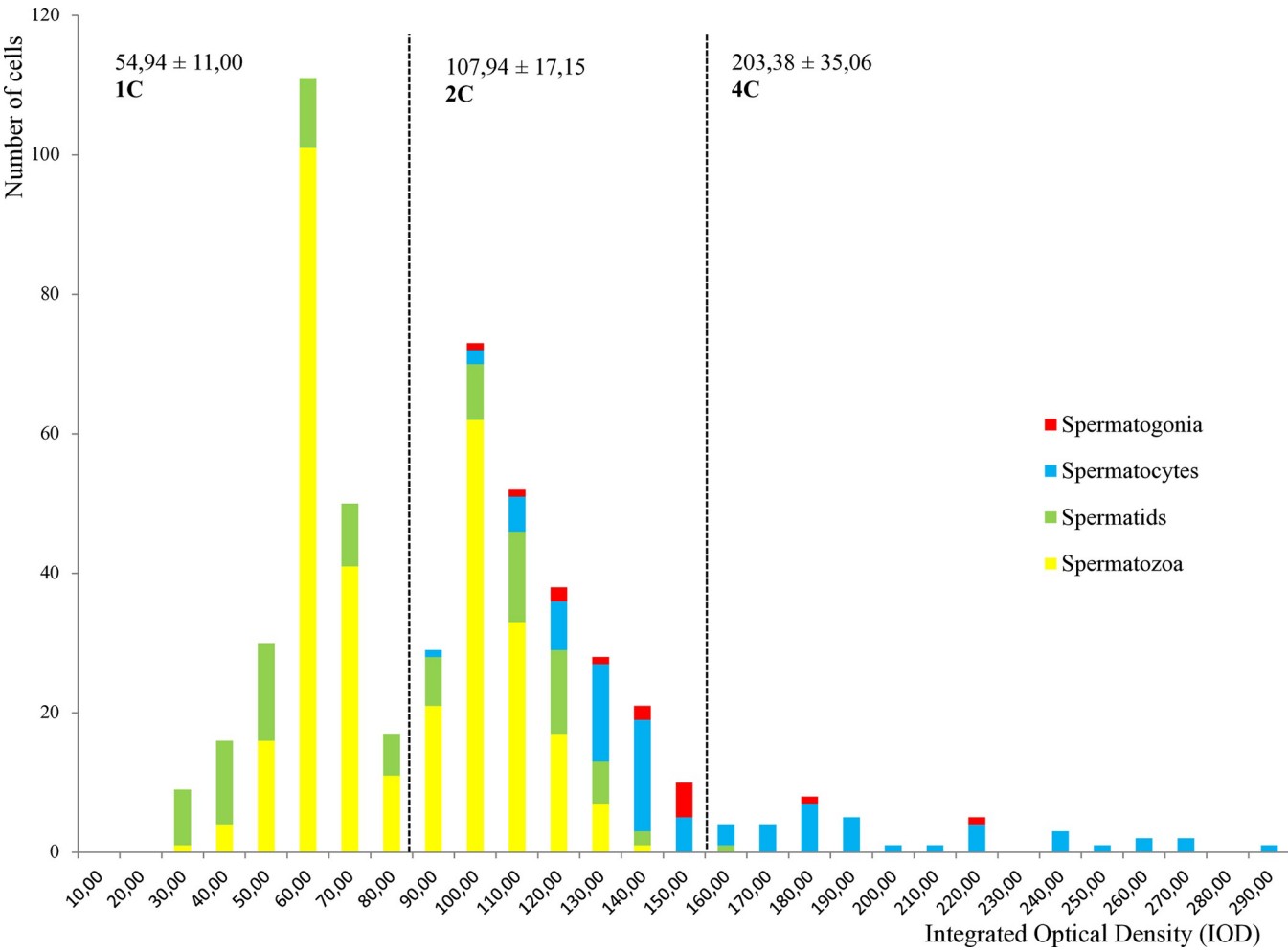

**Fig 4. DNA content variation during spermatogenesis of sexual *C. sandai* (n = 8).** Total number of cells analyzed: 521. 1C (n = 233), 2C (n = 251), 4C (n = 37) cells, where C is the DNA content expressed as the integrated optical density (IOD) given in the X-axis. For each category, the average IOD ± standard deviation is provided. The number of cells is given on the Y-axis.

*japonica*, the distribution of somatic cells of *C.* sp. form A/R could be delineated arbitrarily into five cell populations following their DNA content (Fig 5A, n = 3). Only 3.47% (7/202) of all somatic cells are found to be 1C or less. The second population includes 2C cells (12.38% of cells, 25/202) while most of the somatic cells (70.30%, 142/202) appear as 3C, suggesting triploid individuals as expected for this lineage, confirming the validity of cell populations. Finally, 10.89% (22/202) and 2.97% (6/202) of cells belong to a 4C and a 6C population, respectively. These same cell populations could be retrieved within the distribution of *C.* sp. form A/R male germ cells (Fig 5B, n = 9), indicating that the sperm cells are unreduced. The 1C population constituted only around 1.99% (5/251) of all germ cells and included exclusively spermatids. Three distinct germ cell types were found in the 2C population (9.96% of all cells, 25/251) while most germ cells (61.75% of cells, 155/251) of all types belonged to the 3C population. The 4C population comprised spermatozoa, spermatids, and spermatocytes (19.92% of the cells, 50/251); the 6C population encompassed 6.37% of the cells (16/251) which were spermatids, spermatocytes and spermatogonia. Finally, we observed a unique 8C cell (0.40% of the cells), a spermatogonia.

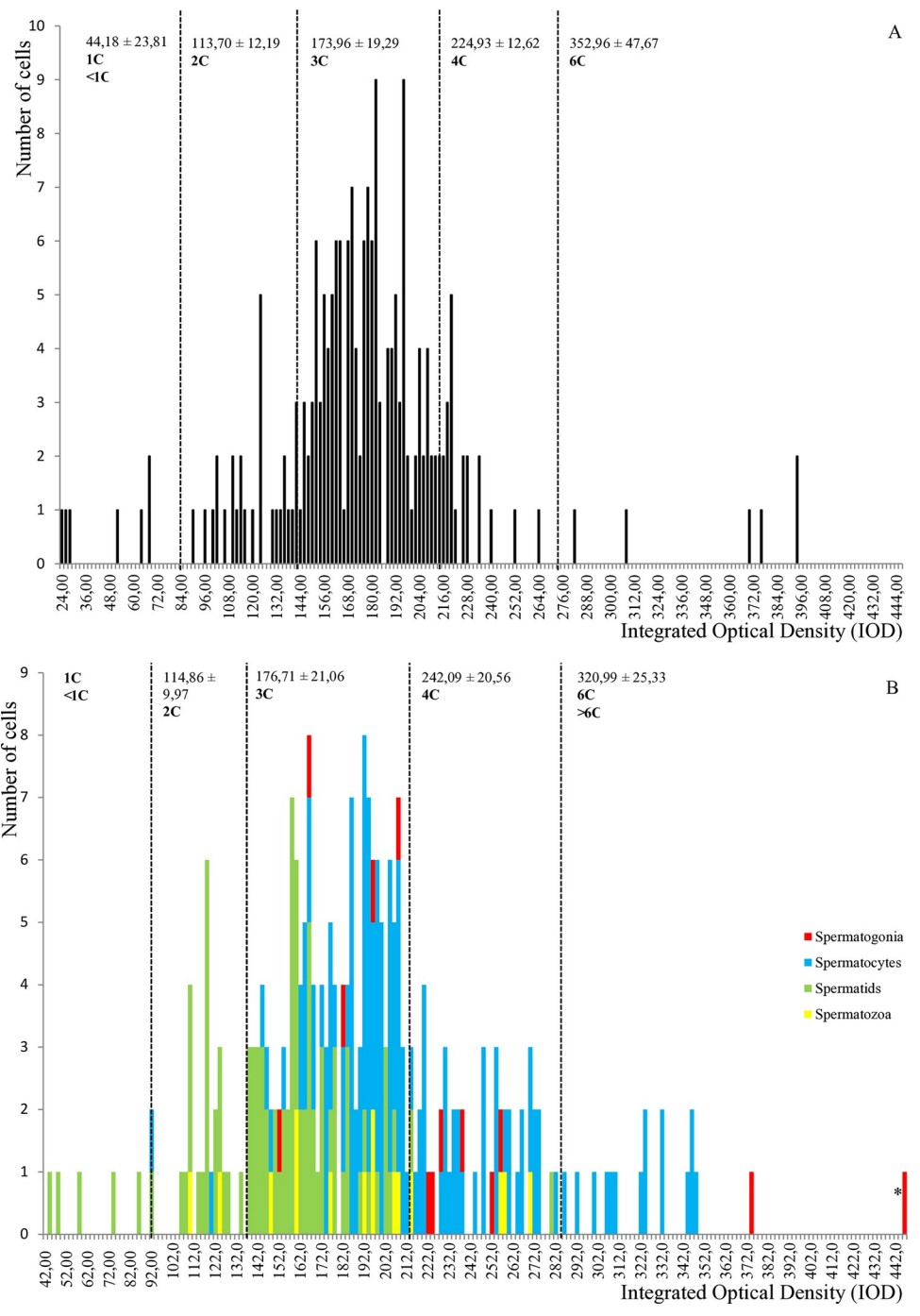

**Fig 5. DNA content in androgenetic *C*. sp. form A/R. A.** DNA content of somatic cells (n = 3). Total number of cells analyzed: 202. 1C and less (n = 7), 2C (n = 25), 3C (n = 142), 4C (n = 22), 6C (n = 6) cells where C is the genetic content. **B.** DNA content of male germ cells (n = 9). Total number of cells analyzed: 251. 1C (n = 5), 2C (n = 25), 3C (n = 155), 4C (n = 50), 6C and more (n = 16), 8C (n = 1) cells, where C is the genetic content. For each category, the average IOD ± standard deviation is provided, except when the number of cells is inferior to six.

## Discussion

### Validation of the image cytometry method

While computer-assisted image cytometry is widely used in cancer research and oncology to study ploidy and morphology changes in cancer cells [60], it is rarely used to study spermatogenesis and meiotic processes, and has, to our knowledge, never been applied to mollusks. The method has however been useful to characterize spermatogenesis in human male infertility [64] and in copepods [65]. The method also revealed the occurrence of meiosis in the coffee rust *Hemileia vastatrix*, which was up to then thought to be clonal [66].

The DNA content distribution of *C. japonica* male germ cells matches with the theoretical distribution of a regular meiosis (Figs 1 and 3). It shows a reduced haploid (1C) population of spermatozoa and spermatids, which are the reduced stages of a traditional spermatogenesis cycle with a regular two-step meiotic division. The 2C population represents G1 gamete cells and cells resulting from the first meiotic division. While spermatogonia are only found to be 2C, spermatocytes are mainly 2C and 4C. This suggests that G2 cells, the replicated gametes in *C. japonica*, are spermatocytes. The absence of 4C spermatogonia in our study, which are normally linked to gonial proliferation, is probably the consequence of their similar nuclear morphology to spermatocytes and the difficulty in distinguishing them. Only early spermatogonia, which are less dense, are easily distinguishable in our study. The replicated spermatogonia may represent a transient stage of spermatogenesis that has been missed here or been categorized as spermatocytes. Our results are in accordance with the described spermatogenesis and sexual cycle of *C. japonica* [67], therefore validating the use of image cytometry to investigate the cytological basis of unreduced sperm formation in *Corbicula*. The percentage of unreduced sperm observed in *C. japonica* is low and could be due to natural errors of meiosis as observed in humans [68]. This is also in accordance with Rybalkina *et al.* [67], which reported a few biflagellate sperm in *C. japonica* (not quantified). Technical and cell type assignment errors to explain the low percentage of unreduced sperm is unlikely since *Corbicula* slender-headed spermatozoa are typical and easily distinguishable. Moreover, all uneasily classifiable cells were considered as debris in our study.

### Cytological mechanisms of spermatogenesis in androgenetic Corbicula clams

Our results suggest that male germ cells of hermaphroditic *C.* sp. from A/R do not perform a regular two-step meiosis with a reductional process, but rather divide through a modified meiosis (Fig 1B). Indeed, to explain the production of unreduced spermatozoa, the endoreplication hypothesis (Fig 1C) can be discarded since no 12C gametes were observed, which would result from an endoreplication of a 3C base cell population. Furthermore, as only very few (2%) reduced spermatids and no reduced spermatozoa were observed in *Corbicula* form A/R, the hypothesis of a canonical meiosis followed by fusion of nuclei (Fig 1D) can also be rejected. The occurrence of a modified meiosis is therefore the cytological mechanism underlying androgenetic sperm formation in *Corbicula* clams.

If the first meiotic division is modified, and homologous chromosomes do not segregate into two distinct cells, the gametes formed are unreduced and retain their heterozygosity along the chromosomes, except where recombination took place, as observed recently in bdelloid rotifer *Adineta vaga* [69]. If the second meiosis is incomplete, and the sister chromatids do not segregate, the gametes formed are unreduced but became homozygous, except where crossing-over took place. As androgenetic *Corbicula* lineages show a high level of heterozygosity [34, 46], an abortive meiosis I is more likely than an abortive meiosis II to produce unreduced

sperm in *Corbicula* clams. The abortion of the second meiotic division is thought to be rare, resulting inexorably to extinction since the masking of recessive deleterious mutations is strongly reduced, unless frequent recombination events occur [3]. Moreover, the occurrence of a functional meiosis I with an odd chromosome number would likely result in aneuploidy. While triploidy is frequent in *Corbicula*, aneuploidy was never observed [70–73]. However, abortion of the second meiotic division was shown in few plant species and resulted in a reduced heterozygosity [51].

During oogenesis of androgenetic *Corbicula* lineages, all the nuclear chromosomes of the oocyte are expulsed as two polar bodies after fertilization during meiosis I [17, 43, 74] instead of a single polar body as in canonical meiosis. These results show that meiosis I is also distinct during *Corbicula* oogenesis in androgenetic lineages and that it seems modified both during spermatogenesis and oogenesis. Meiosis II was not observed during oogenesis in androgenetic *Corbicula* [43] since all chromosomes are discarded during meiosis I, while all chromosomes appear to be conserved in meiosis I during spermatogenesis in androgenetic *Corbicula*. These two processes, the maternal chromosome expulsion and the production of unreduced spermatozoa, both critical in androgenetic lineages, might have a common origin. The maternal chromosome expulsion was already hypothesized as a consequence of centrosome and/or cytoskeleton oddities, with the spindle axis being parallel to the cortex of the oocyte instead of perpendicular, allowing the elimination of all maternal chromosomes as two polar bodies during meiosis I [74, 75]. The meiotic modification causing the production of unreduced spermatozoa could also be explained by such centrosome and/or cytoskeleton oddities, retaining all homologous chromosomes in the spermatocyte during meiosis I. The exact cytological mechanism of this still has to be elucidated and confirmed in the light of the sex-determining system, which is still unknown in *Corbicula*.

Cell population definition during spermatogenesis in *C.* sp. form A/R was less clear than with *C. japonica* data, but the DNA content distribution of the male germ cells was similar to the somatic cells, mostly ranging between 2C and 4C, with spermatids and spermatozoa being detected in the three categories. The distribution of both somatic and germline cells of *C.* sp. form A/R across distinct ploidies (2C, 3C and 4C) is explained by the mixing of different individuals (of different ploidy) for cell suspension preparation due to the low cell density obtained per individual. Among androgenetic *Corbicula*, the three ploidies found here (2n, 3n and 4n) have been described previously with triploid individuals being predominant [43, 70–73, 76]. Individuals with different ploidy levels can also co-exist within the same population [77, 78]. Our data can however not rule out mosaicism (*i.e.*, tissue with different ploidy levels within an individual) in *Corbicula* clams, as both foot and mantle tissues were mixed as somatic controls. Bivalves seem indeed particularly tolerant to such ploidy oddity being found in natural populations [79] or induced for commercial purposes [80], or the result of degraded environmental conditions [81].

## Transition to androgenesis

In the presumably sexual *C. sandai*, about half of the male germ cells seem to go through a canonical meiotic cycle (Fig 1A) comprising 1C, 2C and replicating 4C male gamete cells (Fig 4). A high proportion of spermatozoa and spermatids (44.76 and 33.33% respectively) were nevertheless unreduced (2C) showing for the first time a high percentage of unreduced spermatozoa in a supposedly sexual *Corbicula* lineage. The presence of both reduced and unreduced spermatozoa in *C. sandai* does not allow to rule out the existence of a post-meiotic mechanism restoring the somatic ploidy after a canonical meiosis (Fig 1D). Given the occurrence of a modified meiosis (Fig 1B) in androgenetic *Corbicula*, as shown in the present study,

this process seems a more parsimonious explanation to the production of unreduced sperm in *C. sandai* than the coexistence of two distinct mechanisms within this genus. The histological study of *C. sandai* individuals however confirmed dioecy. Our study therefore suggests that males of *C. sandai* are capable to produce both reduced and unreduced sperm in an almost 50:50 proportion. This assumption seems consistent with sperm morphology, as both mono- and biflagellate sperm have been observed for *C. sandai* individuals (not quantified; Vastrade, personal observation). Since all three species analyzed in this study were processed separately, it is very unlikely that a contamination issue from form A/R to *C. sandai* occurred. Moreover, the 6C cell population detected in form A/R was not retrieved in *C. sandai*, neither in *C. japonica*, excluding a contamination problem.

In Lake Biwa (Japan), where androgenetic individuals co-occur with *C. sandai*, both the sexual and asexual native populations show a high genetic diversity [34, 46]. Remarkably, the three main species described in this lake and river system (*C. sandai*, *C. leana* and *C. fluminea*) shared alleles and therefore formed a single allelic pool [44–46]. These findings advocate for repeated origins of androgenetic lineages in Lake Biwa, from *C. sandai* or one of its ancestors [34, 46], which is confirmed here. As *C. sandai* individuals were pooled in this study, we cannot conclude whether these two types of spermatozoa are produced by distinct *C. sandai* males or not, but preliminary results based on sperm morphology (size and number of flagella) suggest the two types could be produced by the same individual (Vastrade, personal observation). The high proportion of unreduced sperm in the current *C. sandai* population suggests that androgenetic individuals or lineages could be continuously produced.

Repeated origins of asexual taxa from sexual ones is possible in mixed systems where both coexist [*e.g.*, [82–85]], as it is the case in Lake Biwa. In most obligately sexual species, a transition to asexuality would require at least the production of unreduced gametes and their spontaneous development without fertilization [86]. This latter feature is nonetheless not required in the case of androgenesis, where fertilization usually occurs [6, 33]. In *Corbicula*, the two main androgenetic features are: i) the production of unreduced spermatozoa and ii) the maternal genome loss. As discussed previously, the emergence of these two features could be linked. Moreover, the spontaneous production of unreduced gametes, at least in low proportion, is not rare in animals and plants and can even be triggered under some conditions [47, 87, 88]. This was also observed in the strictly sexual species *C. japonica* [[67], present study]. The transition to androgenesis in *Corbicula* clams could therefore start with an increase of unreduced sperm production, as observed in *C. sandai*. The putative genetic bases of the shift from sexuality to androgenesis remain nevertheless unknown in *Corbicula*.

Unique biological features, very different from *C. sandai*, characterize androgenetic *Corbicula* clams. They are hermaphrodites and produce biflagellate unreduced sperm while *C. sandai* is dioecious with reduced and unreduced sperm [[35], present study]. Androgenetic *Corbicula* show an intrabranchial incubation of the juveniles while *C. sandai* is oviparous [reviewed in [35]]. The transition to androgenesis from *C. sandai* is therefore not evident. However, the oviparity of *C. sandai* is based only on a few ancient observations [89] and the brooding capacity of *C. sandai* should be re-assessed [35]. Moreover, the sperm morphology of *C. sandai* from Lake Biwa should be further verified, particularly in males producing unreduced sperm. Males producing unreduced sperm were already detected in a river connected to Lake Biwa; they were however described as androgenetic *C. leana* [78]. Moreover, previous results [44, 46] suggest that *C. sandai* and *C. leana* could belong to the same species. A more thorough study, using genomics and cytology, would be required to understand the relationships between the sexual and androgenetic *Corbicula* species described in Lake Biwa and surrounding rivers.

In general, *Corbicula* clams are characterized by a wide range of reproductive features (sexuality *vs.* androgenesis, hermaphroditism *vs.* dioecy, various brooding strategies, etc.) making it a suitable model to study the evolution of different reproductive characteristics [35].

## Origins, mechanisms, and evolutionary implications of unreduced gamete production

This study showed the occurrence of a modified meiotic pathway in *Corbicula* leading to the production of unreduced spermatozoa. Most known androgenetic systems also imply unreduced male gamete production. In the androgenetic conifer *C. dupreziana*, a spindle system failure during meiosis was hypothesized to explain unreduced pollen production [90]. The production of unreduced sperm in the androgenetic stick insect *Bacillus* was hypothesized via a premeiotic endomitotic chromosome doubling [25], or via a modified meiosis in the androgenetic stick insect *Clonopsis gallica* [91].

Not only androgenetic species produce unreduced gametes, but also parthenogenetic systems where the females produced unreduced oocytes. A modified meiosis (Fig 1B) is a major mechanism to unreduced male or female gamete production in plants [51, 88, 92, 93], with abnormal first and second meiotic divisions detected in at least 13 and 15 taxa, respectively [51]. However, premeiotic (Fig 1C) and post-meiotic (Fig 1D) pathways have also been documented in several plant species [51, 88]. These mechanisms–meiotic, pre-meiotic and post-meiotic modifications–were also described in Metazoans. The bdelloid rotifer *Adineta vaga* [69], the water flea *Daphnia pulex* [94], and the triploid fish *Carassius gibelio* [95] are notable examples of unrelated taxa producing unreduced oocytes via suppression of meiosis I. The gynogenetic loach *Misgurnus anguillicaudatus* produces unreduced eggs via a pre-meiotic endomitosis [96]. Moreover, the production of unreduced sperm via a post-meiotic pathway was successfully induced in this species, following a mitosis-controlling gene suppression (Zhang *et al.* 2023). Noteworthy, elucidating the mechanisms and origins of unreduced gametes is often difficult, notably because distinct meiotic abnormalities can co-occur in a single species or even individual [51, 92].

Many factors can induce or increase unreduced gamete production. In most studies, external factors–such as temperature or pressure variations, herbivory or pathogen attacks, food or water shortages–were shown to impact unreduced gamete production in plants [51, 88] but also in amphibians and fishes [87, 97, 98]. Notably, heat stress during meiosis was shown to induce or increase the frequency of failure of chromosome segregation or spindle disorientation [99]. Species that mate in unstable environments are thus predicted to be more prone to environment-driven unreduced gamete production [87]. However, the impact of environmental factors on *Corbicula* gamete production was never assessed. Mutations in meiosis or mitosis-controlling genes can also lead to the production of unreduced gametes in plants [51, 88, 92, 93, 100] and in animals [95, 101].

As a result, unreduced gametes are almost universal: they are found at least in plants, insects, fishes, amphibians, and even in mammals and birds, where they are however lethal [reviewed in [47]]. Their ubiquity, heritability, and diverse production mechanisms suggest they may offer evolutionary advantages [47]. First, polyploidy speciation, which help prevent species extinction, could be facilitated by unreduced gamete production. This is particularly relevant in asexual organisms, such as *Corbicula* clams, where polyploidy enables reproduction despite disrupted meiosis [3]. Polyploidy also compensate for asexual reproduction disadvantages such as preventing the loss of masking recessive deleterious mutations [3]. Second, the enhanced ability to produce unreduced gametes in response to stress suggest that polyploid speciation would be selected as a mean to escape disturbed habitats [47]. Therefore, unreduced

gamete production is crucial in asexual organisms, particularly in hostile and changing environments, as it confers adaptive potential through polyploid formation [see [102, 103] for reviews on polyploidy consequences].

## Conclusion

This study significantly contributed in lifting the veil on the cytological process underlying spermatogenesis in androgenetic and sexual *Corbicula* clams. We bring here cytological evidence of a modified meiosis during spermatogenesis in these organisms. The present results, combined with previously collected data, suggest an abortive first meiotic division. We also demonstrated that the previously acknowledged sexual *C. sandai* produces a high proportion of unreduced spermatozoa, suggesting that this lineage may be at the origin of androgenesis in *Corbicula* and continuously contribute to androgenetic *Corbicula* lineages.

## Supporting information

**S1 Fig. Progression of gametogenesis throughout the year in mature hermaphroditic *Corbicula* sp. form A/R individuals (n = 60). A.** Percentage of female and male follicles. Mixed regions of the tubules were omitted for more clarity. **B.** Spermatogenesis cycle. The tubule stages refer to S1 Table. For each spermatogenesis stage, the number of observations varies between 0 (not observed) and 5 (observed in all individuals).
(DOCX)

**S1 File. Feulgen staining protocol.**
(DOCX)

**S1 Table. Spermatogenesis stages description of *Corbicula* sp. form A/R tubules.**
(DOCX)

## Acknowledgments

The authors gratefully thank V. Deglas (LabCéTi, URPhyM, UNamur), D. Van Vlaender (LabCéTi, URPhyM, UNamur), C. Didembourg (URBM, UNamur) and B. Adeline (UMR BOREA, Caen) for technical advice for both histological techniques and flow cytometry. For help in sampling *C. sp*. form A/R and in culturing collected individuals, we would also like to thank E. Falisse, J. Marescaux and J. Lorquet (URBE, UNamur). Finally, we are thankful to Professors N. Yasuda and M. Yamaguchi (University of Miyazaki) for their precious help in collecting sexual *Corbicula* as well as to the Seta Fishery cooperative (Ohtsu City, Shiga, Japan).

## Author Contributions

**Conceptualization:** Emilie Etoundi, Karine Van Doninck.

**Data curation:** Emilie Etoundi, Martin Vastrade, Clothilde Berthelin.

**Formal analysis:** Emilie Etoundi, Martin Vastrade, Clothilde Berthelin, Kristell Kellner, Mélanie Fafin-Lefèvre.

**Funding acquisition:** Karine Van Doninck.

**Investigation:** Emilie Etoundi.

**Methodology:** Emilie Etoundi, Kristell Kellner, Mélanie Fafin-Lefèvre.

**Software:** Mélanie Fafin-Lefèvre.

**Supervision:** Karine Van Doninck.

**Validation:** Emilie Etoundi, Martin Vastrade, Clothilde Berthelin.

**Writing – original draft:** Emilie Etoundi, Karine Van Doninck.

**Writing – review & editing:** Martin Vastrade, Clothilde Berthelin, Kristell Kellner, Mélanie Fafin-Lefèvre, Karine Van Doninck.

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
