## [Decision Letter · Decision Letter 0]

16 Sep 2024

PONE-D-24-33293Transition from sexuality to androgenesis through a meiotic modification in freshwater Corbicula clamsPLOS ONE

Dear Dr. VASTRADE,

Thank you for submitting your manuscript to PLOS ONE. After careful consideration, we feel that it has merit but does not fully meet PLOS ONE’s publication criteria as it currently stands. Therefore, we invite you to submit a revised version of the manuscript that addresses the points raised during the review process.

We look forward to receiving your revised manuscript.

Kind regards,

Qinghua Shi

Academic Editor

PLOS ONE

Journal Requirements: When submitting your revision, we need you to address these additional requirements. 1. Please ensure that your manuscript meets PLOS ONE's style requirements, including those for file naming. The PLOS ONE style templates can be found at https://journals.plos.org/plosone/s/file?id=wjVg/PLOSOne_formatting_sample_main_body.pdf and https://journals.plos.org/plosone/s/file?id=ba62/PLOSOne_formatting_sample_title_authors_affiliations.pdf 2. In your Methods section, please provide additional information regarding the permits you obtained for the work. Please ensure you have included the full name of the authority that approved the field site access and, if no permits were required, a brief statement explaining why. 3. Thank you for stating the following financial disclosure: "The present study has been funded by the University of Namur and the National Funds for Scientific Research (FNRS) through a PhD thesis grant (FC91712) to E.E. and by a PDR research grant (CARMA, 14596412) to K.V.D. " Please state what role the funders took in the study.  If the funders had no role, please state: ""The funders had no role in study design, data collection and analysis, decision to publish, or preparation of the manuscript."" If this statement is not correct you must amend it as needed. Please include this amended Role of Funder statement in your cover letter; we will change the online submission form on your behalf. 4. Please review your reference list to ensure that it is complete and correct. If you have cited papers that have been retracted, please include the rationale for doing so in the manuscript text, or remove these references and replace them with relevant current references. Any changes to the reference list should be mentioned in the rebuttal letter that accompanies your revised manuscript. If you need to cite a retracted article, indicate the article’s retracted status in the References list and also include a citation and full reference for the retraction notice.

Reviewers' comments:

Reviewer's Responses to Questions

**Comments to the Author**

1. Is the manuscript technically sound, and do the data support the conclusions?

Reviewer #1: Yes

Reviewer #2: Yes

2. Has the statistical analysis been performed appropriately and rigorously? 

Reviewer #1: N/A

Reviewer #2: Yes

3. Have the authors made all data underlying the findings in their manuscript fully available?

Reviewer #1: Yes

Reviewer #2: Yes

4. Is the manuscript presented in an intelligible fashion and written in standard English?

Reviewer #1: Yes

Reviewer #2: Yes

5. Review Comments to the Author

Reviewer #1: In their paper, Etoundi and colleagues present the first study evaluating the potential mechanism of spermatogenesis in obligate androgenetic species. The main objectives of the paper are to 1) understand how to produce unreduced gametes through spermatogenesis and 2) understand why androgenesis is repeatedly found in Corbicula clams. To achieve this, they successfully employed a computer-assisted image cytometry technique in a non-model organism, the Corbicula clam. They used image cytometry to distinguish among three mechanisms (abortive step of meiosis, fusion, or chromosome duplication). They demonstrated that:

- C. sandai and C. japonica :

o Both sexual species, C. japonica and C. sandai, are diploid and mainly produce reduced spermatozoa.

o Natural sexual species produce unreduced spermatozoa, with C. sandai representing 44.76 and 33.33% of unreduced spermatozoa and spermatids. This indicates a permissive capacity for reproducing asexually. The modification of meiosis leading to the formation of these unreduced is not identified, but the data excluded pre-meiotic endoduplication.

o In both C. sandai and C. japonica, the existence of unreduced spermatozoa correlates with the observation of biflagellate spermatozoa.

- Corbicula form A/R:

o The data shows different amounts of genetic material in somatic tissue (1C, 2C, 3C, 4C and 6C), indicating distinct levels of ploidy. These different ploidy states could belong to the same individual or represent variations in ploidy among individuals of the same species.

o The absence of 12C content clearly excludes pre-meiotic duplication during spermatogenesis.

o Furthermore, due to the high level of heterozygosity, abortive meiosis I is a more parsimonious mechanism than abortive meiosis II.

Minor comments:

- Were you able to distinguish the 2C biflagellate from the two overlapping 1C spermatozoa, or are they considered debris?

- What is the sex-determining system of Corbicula form A/R? Could it confirm your results, i.e., abortive meiosis I?

- Do you observe embryonic lethality?

- in Corbicula form A/R:

o In the case of triploid chromosomes where pre-meiotic endoreplication does not occur, homologous chromosomes are unable to pair. Consequently, incomplete meiosis I is inevitable, or if meiosis is attempted, it will lead to incorrect segregation of the chromosomes.

o Is it possible that the observed 2C and 1C spermatozoa resulted from inequivalent segregation of homologous chromosomes during meiosis I, producing two products with either 2C or 1C content?

o The somatic presents different ploidy states that could result from multiple fertilisations, leading to the development of mosaic animals.

Reviewer #2: The manuscript by Etoundi & al. analyses the origin of unreduced sperm production in an androgenetic Corbicula species. Androgenesis is a fascinating reproductive system in which the oocyte DNA is completely expelled at fertilisation. As a result, the zygote develops only from paternal DNA, maintaining diploidy if the sperm is unreduced.

Previous observations have shown the presence of biflagellate spermatozoa in an androgenetic Corbicula. Since the flagella is derived from a centrosome, which is inherited along with a nucleus in each sperm cell at the end of meiosis, the presence of two flagella per sperm cell suggests that the nuclei should also be present in two copies in the sperm cell.

However, this has not been verified, and the present study aims to confirm the DNA content of mature spermatozoa in androgenetic Corbicula species compared to normally sexual species.

The study is interesting and new. I have however a couple of important point to raise and several minor comments.

Major comments:

- It is necessary to give a clear explanation of the calibration of ploidy in this study. It is clear for the sexual species where the reduced spermatozoa can be directly assigned to 1C. What defines 1C in the androgenetic species? The rare cells with the lowest DNA content? The whole study and analysis is based on this, it is necessary to clarify this point.

- The androgenetic species studied does not originate from Lake Biwa, but from a river in Belgium. This is not explicitly stated in the main text, but only in the materials and methods. Although it doesn't change the conclusion of the study, it does change the discussion a bit, because since we don't know the origin of this Belgian species, it is more difficult to imagine that C. sandai serves as a reservoir for new androgenetic species. Please explain why the androgenetic species from Lake Biwa mentioned in the introduction has not been studied here.

- Please explicitly explain the link between biflagellate and unreduced sperm.

- Conclusion of unreduced gamete being also found in C. sandai: could it come from a mistake in the assignment of cell types? For instance if spermatocytes and some spermatids are mixed up ? this should be included in the discussion as a possibility

- The discussion takes too long

-

Minor comments:

- include Corbicula in the keywords

-abstract: “by the production unreduced sperm”, OF is missing

-abstract: Rather than "a potential switch between sexuality and androgenesis", would you say that “the production of unreduced sperm in C. sandai may condition or provide the potential for the emergence of androgenesis”?

-line 75: out of curiosity, in stick insects, if the system produces only males, does it mean there is fusion between X-bearing and Y-bearing sperm only ?

-line 97, please specify if these are self-fertilizing hermaphrodites or cross-fertilizing

-line 109: not entirely clear because the defect in female meiosis does not lead to fusion of meiotic products: “as observed during oogenesis” is a bit misleading

-line 159, a reference for the culture conditions is missing

-line 177: Can you explain why you use the Feulgen stain and not the classic Dapi stain?

-line 236: more abundant than ? than male tissue ?

-Figure 1: specify this is the theoretical expectation, not a result of this study (this could go in a supplementary figure, not as a main figure)

-Figure 2: The panel showing the SR region is of poor quality, this is a very important part of the study, this needs to be improved.

-line 448: “these mechanisms” refer to what. ?pre and postmeiotic defects as described in plants ? the transition is unclear

-line 468: “suggest an evolutionary utility for such unsual gametes” ..but it could also simply reflect that meiosis is not perfect and often fails ..(see the many meiotic defects in female meiosis in mouse and humans for instance)

-line 468-471: improve, not entirely clear

-line 479: the sentence is a bit unclear: “unreduced gametes in asexual confer an adaptative potential”: unreduced gametes confer an adaptative potential for the emergence of asexuality (because this is a prerequisite) ..but for an established asexual lineage, I don’t see the adaptative advantage.

-line 481: there have been other cytological studies prior to this study, in particular those that have described the biflagellate sperm in Corbicula.

6. PLOS authors have the option to publish the peer review history of their article (what does this mean?). If published, this will include your full peer review and any attached files.

Reviewer #1: No

Reviewer #2: No

---

## [Author Response · Author response to Decision Letter 0]

4 Oct 2024

We are thankful to the editor and reviewers for their careful revision of our manuscript. The present response was also submitted as a letter, with the reviewer’s comments displayed in italic and the authors’ responses in bold for a better legibility. We wish to draw your attention on a small title modification: “Transition from sexuality to androgenesis through a meiotic modification in freshwater Corbicula clams” is now “Transition from sexuality to androgenesis through a meiotic modification during spermatogenesis in freshwater Corbicula clams”. Note that the line numbers in this document refer to the revised manuscript with track changes and might be slightly different in the unmarked version.

Reviewer #1: In their paper, Etoundi and colleagues present the first study evaluating the potential mechanism of spermatogenesis in obligate androgenetic species. The main objectives of the paper are to 1) understand how to produce unreduced gametes through spermatogenesis and 2) understand why androgenesis is repeatedly found in Corbicula clams. To achieve this, they successfully employed a computer-assisted image cytometry technique in a non-model organism, the Corbicula clam. They used image cytometry to distinguish among three mechanisms (abortive step of meiosis, fusion, or chromosome duplication). They demonstrated that:

- C. sandai and C. japonica :

o Both sexual species, C. japonica and C. sandai, are diploid and mainly produce reduced spermatozoa.

o Natural sexual species produce unreduced spermatozoa, with C. sandai representing 44.76 and 33.33% of unreduced spermatozoa and spermatids. This indicates a permissive capacity for reproducing asexually. The modification of meiosis leading to the formation of these unreduced is not identified, but the data excluded pre-meiotic endoduplication.

o In both C. sandai and C. japonica, the existence of unreduced spermatozoa correlates with the observation of biflagellate spermatozoa.

- Corbicula form A/R:

o The data shows different amounts of genetic material in somatic tissue (1C, 2C, 3C, 4C and 6C), indicating distinct levels of ploidy. These different ploidy states could belong to the same individual or represent variations in ploidy among individuals of the same species.

o The absence of 12C content clearly excludes pre-meiotic duplication during spermatogenesis.

o Furthermore, due to the high level of heterozygosity, abortive meiosis I is a more parsimonious mechanism than abortive meiosis II.

Minor comments:

- Were you able to distinguish the 2C biflagellate from the two overlapping 1C spermatozoa, or are they considered debris?

Unfortunately, the Feulgen approach did not allow distinguishing the flagella, that are particularly thin. This staining is well-adapted for DNA staining but makes other structures hardly visible. 

- What is the sex-determining system of Corbicula form A/R? Could it confirm your results, i.e., abortive meiosis I?

Thank you for asking this question. Bivalve sex-determining mechanisms were quite recently reviewed (here: https://doi.org/10.1007/978-3-319-94139-4_6), with a short discussion on Corbicula androgenesis. Unfortunately, the sex-determining system in Corbicula is still unknown, and because our discussion section is already long (as noticed by reviewer #2), we have chosen not to develop a discussion on sex determination in our manuscript. However, this is still a good point to be mentioned. We have added a short sentence on this topic in lines 375-376.

- Do you observe embryonic lethality?

This is a good question. To our knowledge, this was never studied in Corbicula. This might be considered for future studies.

- in Corbicula form A/R:

o In the case of triploid chromosomes where pre-meiotic endoreplication does not occur, homologous chromosomes are unable to pair. Consequently, incomplete meiosis I is inevitable, or if meiosis is attempted, it will lead to incorrect segregation of the chromosomes.

This is a good remark. Indeed, meiosis I with an odd chromosome number would likely result in aneuploidy, which was never observed in Corbicula clams. This was discussed lines 356-358.

o Is it possible that the observed 2C and 1C spermatozoa resulted from inequivalent segregation of homologous chromosomes during meiosis I, producing two products with either 2C or 1C content?

We formulated this hypothesis at some point, but it is likely not verified, so we have chosen not to include it in the manuscript. Mixed production of both 1C and 2C spermatozoa in high proportions was only observed in C. sandai, who is diploid. Moreover, inequivalent segregation of homologous chromosomes during meiosis I would most likely produce aneuploid products, except if a cytological mechanism allows one sister cell to receive only one chromosome from each set and the other sister cell to receive always two chromosomes from each set. Such mechanism was never demonstrated in Corbicula.

o The somatic presents different ploidy states that could result from multiple fertilisations, leading to the development of mosaic animals.

Mosaicism is indeed not excluded in Corbicula clams, as discussed lines 385-389. However, these different ploidy states might also result from different individuals, as ploidy is variable in these clams, even within the same species.

Reviewer #2: The manuscript by Etoundi & al. analyses the origin of unreduced sperm production in an androgenetic Corbicula species. Androgenesis is a fascinating reproductive system in which the oocyte DNA is completely expelled at fertilisation. As a result, the zygote develops only from paternal DNA, maintaining diploidy if the sperm is unreduced.

Previous observations have shown the presence of biflagellate spermatozoa in an androgenetic Corbicula. Since the flagella is derived from a centrosome, which is inherited along with a nucleus in each sperm cell at the end of meiosis, the presence of two flagella per sperm cell suggests that the nuclei should also be present in two copies in the sperm cell.

However, this has not been verified, and the present study aims to confirm the DNA content of mature spermatozoa in androgenetic Corbicula species compared to normally sexual species.

The study is interesting and new. I have however a couple of important point to raise and several minor comments.

Major comments:

- It is necessary to give a clear explanation of the calibration of ploidy in this study. It is clear for the sexual species where the reduced spermatozoa can be directly assigned to 1C. What defines 1C in the androgenetic species? The rare cells with the lowest DNA content? The whole study and analysis is based on this, it is necessary to clarify this point.

We agree that distinct ploidy populations could not be retrieved as clearly for form A/R than for C. sandai and C. japonica (lines 299-300) and that the delimitations of the different cell populations based on the cell distribution is a bit arbitrary (line 301). However, a comparison between somatic and sperm cells in form A/R clearly show similar profiles, indicating that the sperm cells are unreduced. Moreover, this lineage is known for being mostly triploid, and our analysis of the cell distribution is consistent with this prior knowledge.

- The androgenetic species studied does not originate from Lake Biwa, but from a river in Belgium. This is not explicitly stated in the main text, but only in the materials and methods. Although it doesn't change the conclusion of the study, it does change the discussion a bit, because since we don't know the origin of this Belgian species, it is more difficult to imagine that C. sandai serves as a reservoir for new androgenetic species. Please explain why the androgenetic species from Lake Biwa mentioned in the introduction has not been studied here.

The original aim of this study was to understand the androgenetic mechanism leading to unreduced sperm production of the most widespread invasive lineages in Belgium, namely C. sp. form A/R. To achieve this, sexual species had to be acquired from another continent, because no sexual species was ever recorded in the invaded area, i.e., Europe and America. Sexual species from Asia were selected notably because the invasion likely originated from this continent. In a recent study (Vastrade et al. 2022, https://peercommunityjournal.org/articles/10.24072/pcjournal.180/), we have shown the occurrence of a substantial allele sharing between European lineages and Lake Biwa species. It is thus not that difficult to imagine that C. sandai serves as a reservoir for new androgenetic species, some of them being introduced in other continents.

- Please explicitly explain the link between biflagellate and unreduced sperm.

Thank you for highlighting that this point was not clear. It was clarified lines 123-125.

- Conclusion of unreduced gamete being also found in C. sandai: could it come from a mistake in the assignment of cell types? For instance if spermatocytes and some spermatids are mixed up ? this should be included in the discussion as a possibility

Such confusion would be unlikely because of the characteristic morphology of mature spermatozoa compared to spermatogonia and spermatocytes, and because any doubtful cell was considered as debris. This is discussed lines 336-338.

- The discussion takes too long

The section that you point as unclear was significantly reduced, lines 477-492. If you detect other sections which would deserve to be reduced, please let us know.

Minor comments:

- include Corbicula in the keywords

Good point, this was added line 39.

-abstract: “by the production unreduced sperm”, OF is missing

Thank you, this was corrected line 27.

-abstract: Rather than "a potential switch between sexuality and androgenesis", would you say that “the production of unreduced sperm in C. sandai may condition or provide the potential for the emergence of androgenesis”?

This is well written, but this suggestion would not reflect the transition from sexuality to androgenesis. We thus adapted it into: “The production of unreduced sperm may condition or provide the potential for the emergence of androgenesis in this sexual species.” See lines 32-34.

-line 75: out of curiosity, in stick insects, if the system produces only males, does it mean there is fusion between X-bearing and Y-bearing sperm only ?

We thank you for your curiosity on androgenesis. The Bacillus stick insect sex-determining system in a X0 system, with XX females and X0 males. On one hand, the all-male androgenesis is likely achieved through fertilization by balanced unreduced spermatozoa. The offspring thus receive all chromosomes from the father, including the sexual chromosomes X0, resulting in an all-male progeny. On the other hand, fusion of two sperms result in a bisexual progeny. Note that this progeny is often biased to female (e.g., 78% females in Tinti & Scali 1995, https://doi.org/10.1002/jez.1402730208), while theory (e.g., Punnett square on sexual chromosomes from two X0 spermatozoa) would predict 67% males (with 00 individuals considered not viable). Genetic alone thus fails explaining the observed prevalence of females in this case, which might be due to selection.

-line 97, please specify if these are self-fertilizing hermaphrodites or cross-fertilizing

This was specified lines 98 and 101.

-line 109: not entirely clear because the defect in female meiosis does not lead to fusion of meiotic products: “as observed during oogenesis” is a bit misleading

The sentence (currently lines 111-113) was subdivided to clarify that the comparison to oogenesis is only about an aborted meiotic division.

-line 159, a reference for the culture conditions is missing

A reference was added lines 164 and 667-669.

-line 177: Can you explain why you use the Feulgen stain and not the classic Dapi stain?

The Feulgen staining offers a stochiometric relationship between DNA content and staining intensity, which can be measured by image analysis with a high precision. This method was already optimized in Fafin-Lefevre et al. 2011 (https://pubmed.ncbi.nlm.nih.gov/21980622/). The DAPI staining is less adapted for this kind of analysis.

-line 236: more abundant than ? than male tissue ?

Indeed, this was specified line 241.

-Figure 1: specify this is the theoretical expectation, not a result of this study (this could go in a supplementary figure, not as a main figure)

This was specified line 788. We have made the choice to leave it as a main figure because it is important for understanding the manuscript. 

-Figure 2: The panel showing the SR region is of poor quality, this is a very important part of the study, this needs to be improved.

The figure was redone.

-line 448: “these mechanisms” refer to what. ?pre and postmeiotic defects as described in plants ? the transition is unclear

This was specified line 457.

-line 468: “suggest an evolutionary utility for such unsual gametes” ..but it could also simply reflect that meiosis is not perfect and often fails ..(see the many meiotic defects in female meiosis in mouse and humans for instance)

This claim was toned down lines 477-478.

-line 468-471: improve, not entirely clear

This section was reworked and reduced lines 477-492.

-line 479: the sentence is a bit unclear: “unreduced gametes in asexual confer an adaptative potential”: unreduced gametes confer an adaptative potential for the emergence of asexuality (because this is a prerequisite) ..but for an established asexual lineage, I don’t see the adaptative advantage.

Thank you for this remark, highlighting that this section was indeed unclear. The adaptive potential can be conferred through polyploidy. We hope this section is clearer now.

-line 481: there have been other cytological studies prior to this study, in particular those that have described the biflagellate sperm in Corbicula.

This claim was toned down line 494.

---

## [Decision Letter · Decision Letter 1]

23 Oct 2024

PONE-D-24-33293R1Transition from sexuality to androgenesis through a meiotic modification during spermatogenesis in freshwater Corbicula clamsPLOS ONE

Dear Dr. VASTRADE,

Thank you for submitting your manuscript to PLOS ONE. After careful consideration, we feel that it has merit but does not fully meet PLOS ONE’s publication criteria as it currently stands. Therefore, we invite you to submit a revised version of the manuscript that addresses the points raised during the review process.

You should revise your manuscript based on reviewers' suggestions, questions and comments, not just answer their questions in the letter. 

We look forward to receiving your revised manuscript.

Kind regards,

Qinghua Shi

Academic Editor

PLOS ONE

Journal Requirements:

Reviewers' comments:

Reviewer's Responses to Questions

**Comments to the Author**

1. If the authors have adequately addressed your comments raised in a previous round of review and you feel that this manuscript is now acceptable for publication, you may indicate that here to bypass the “Comments to the Author” section, enter your conflict of interest statement in the “Confidential to Editor” section, and submit your "Accept" recommendation.

Reviewer #1: All comments have been addressed

Reviewer #2: All comments have been addressed

2. Is the manuscript technically sound, and do the data support the conclusions?

Reviewer #1: Yes

Reviewer #2: Yes

3. Has the statistical analysis been performed appropriately and rigorously? 

Reviewer #1: N/A

Reviewer #2: Yes

4. Have the authors made all data underlying the findings in their manuscript fully available?

Reviewer #1: Yes

Reviewer #2: Yes

5. Is the manuscript presented in an intelligible fashion and written in standard English?

Reviewer #1: Yes

Reviewer #2: Yes

6. Review Comments to the Author

Reviewer #1: (No Response)

Reviewer #2: I am satisfied with all the answers given in the letter. I regret, however, that some answers and clarifications have not been incorporated into the manuscript, in particular :

-explanation on the calibration of ploidy for the A/R form

-the rationale for studying the sexual form in lake Biwa, while the androgenetic species is from Belgium

-the rationale for using the Feulgen staining and not DAPI: simply state somewhere in the text that DAPI is less adapted in this case …

7. PLOS authors have the option to publish the peer review history of their article (what does this mean?). If published, this will include your full peer review and any attached files.

Reviewer #1: No

Reviewer #2: No

---

## [Author Response · Author response to Decision Letter 1]

24 Oct 2024

The clarifications asked were incorporated in the manuscript, lines 159-164; 189; and 308-311.

---

## [Decision Letter · Decision Letter 2]

30 Oct 2024

Transition from sexuality to androgenesis through a meiotic modification during spermatogenesis in freshwater Corbicula clams

PONE-D-24-33293R2

Dear Dr. Martin VASTRADE,

We’re pleased to inform you that your manuscript has been judged scientifically suitable for publication and will be formally accepted for publication once it meets all outstanding technical requirements.

Kind regards,

Qinghua Shi

Academic Editor

PLOS ONE

Additional Editor Comments (optional):

Reviewers' comments:

Reviewer's Responses to Questions

**Comments to the Author**

1. If the authors have adequately addressed your comments raised in a previous round of review and you feel that this manuscript is now acceptable for publication, you may indicate that here to bypass the “Comments to the Author” section, enter your conflict of interest statement in the “Confidential to Editor” section, and submit your "Accept" recommendation.

Reviewer #2: All comments have been addressed

2. Is the manuscript technically sound, and do the data support the conclusions?

Reviewer #2: Yes

3. Has the statistical analysis been performed appropriately and rigorously? 

Reviewer #2: Yes

4. Have the authors made all data underlying the findings in their manuscript fully available?

Reviewer #2: Yes

5. Is the manuscript presented in an intelligible fashion and written in standard English?

Reviewer #2: Yes

6. Review Comments to the Author

Reviewer #2: (No Response)

7. PLOS authors have the option to publish the peer review history of their article (what does this mean?). If published, this will include your full peer review and any attached files.

Reviewer #2: No

---

## [Editor Report · Acceptance letter]

15 Nov 2024

PONE-D-24-33293R2 

PLOS ONE

Dear Dr. VASTRADE, 

I'm pleased to inform you that your manuscript has been deemed suitable for publication in PLOS ONE. Congratulations! Your manuscript is now being handed over to our production team.

Kind regards, 

on behalf of

Professor Qinghua Shi 

Academic Editor

PLOS ONE